# Knowledge, Attitude and Perception of Pharmacy Students towards Pharmacogenomics and Genetics: An Observational Study from King Saud University

**DOI:** 10.3390/genes13020269

**Published:** 2022-01-29

**Authors:** Azher Arafah, Muneeb U. Rehman, Wajid Syed, Salmeen D. Babelghaith, Abdulrahman Alwhaibi, Mohamed N. Al Arifi

**Affiliations:** Department of Clinical Pharmacy, College of Pharmacy, King Saud University, Riyadh 11451, Saudi Arabia; aazher@ksu.edu.sa (A.A.); muneebjh@gmail.com (M.U.R.); wali@ksu.edu.sa (W.S.); sbabelghaith@ksu.edu.sa (S.D.B.); malarifi@ksu.edu.sa (M.N.A.A.)

**Keywords:** pharmacogenomics, knowledge, attitude, pharmacy students, genetic testing

## Abstract

Pharmacists are considered among the most accessible healthcare workers in fundamental positions to implement new clinical initiatives, such as pharmacogenomics services. The scope of pharmacogenomics in improving health outcomes and the quality of health care is well-known. Implementation of such initiatives requires adequate knowledge, perception, and positive attitudes among pharmacists. A study was conducted on pharmacy students at King Saud University in Riyadh to analyze their attitudes, knowledge, and perceptions concerning pharmacogenomics to explore the feasibility of establishing full-time pharmacogenomics instruction and services. A cross-sectional study was carried out in one of the significant pharmacy schools of Saudi Arabia, using a simple questionnaire-based survey in pharmacy students pursuing Bpharm and PharmD courses to obtain preliminary information about pharmacogenomics among the surveyed population. The study’s secondary objective was to determine the perceived belief about pharmacogenomics implementation in clinical practice. Out of the total of 552 participants, 41.8% correctly defined pharmacogenomics and 81.3% understood that genetic change could lead to adverse reactions. More than half of the participants agreed that the FDA recommends pharmacogenomics testing for certain drugs. The knowledge about a year of use of pharmacogenomics in clinical practice was found to be very low; only 15.2% could correctly answer. Only 60% of students agreed on pharmacogenomics testing for selecting the therapy with the most negligible adverse effects. Due to the limited knowledge about and understanding of pharmacogenomics, there is a lack of interest among pharmacy students in implementing pharmacogenomics testing in clinical practice. Our study highlights the need for improving pharmacy students’ knowledge about pharmacogenomics and pharmacogenetics so that the implementation of pharmacogenomics testing in clinical practice will become easier. There is a need to introduce an up-to-date curriculum for pharmacy courses other pharmacogenomics-based health education programs in Saudi Arabia.

## 1. Introduction

Pharmacogenomics deals with the application of genomics in pharmacology to study and analyze the outcome of genetic variations in responses to drug treatments [1]. The study of variability in drug responses due to genetic variations is known as pharmacogenetics [2]. The difference in genes is responsible for the interaction of drugs with their molecular targets, thus affecting the drugs’ efficacy and validating unwanted side effects [2]. As healthcare is progressing forward, the application of pharmacogenomics in the recent past has also increased tremendously, paving the way for tailored medicine as per the genetic map of individuals [3]. With the advent of pharmacogenomics, pharmacogenetics, and the completion of the Human Genome Project, anticipation is high that genetic information would radically improve medicine, that side effects would be more predictable, and that patients could be screened for likely drug responses [4].

As per the pharmacogenomics report, so far, more than 350 drugs have been incorporated in the Food and Drug Administration’s (US-FDA) repository of drugs labelled prior to administration [5]. These mostly include drugs having a thin therapeutic index and that are potentially toxic, e.g., anti-neoplastic, anti-coagulant, and anticonvulsant dugs [6,7,8,9]. The aim of clinical pharmacogenomics testing is to alert physicians to prescribed medications that have known genetic variants that are associated with adverse drug reactions and the effectiveness of the drug [10]. In Saudi Arabia, the prevalence of pharmacogenomics testing CYP2C9*2 and CYP2C9*3 were found with rates of 11% and 9%, respectively [11]. Furthermore, studies have discovered that genetic variation in Saudi Arabia is high for a variety of causes, emphasizing the importance of pharmacists’ education in pharmacogenomics [11,12]. There have been several reports published previously from around the world to study the knowledge or attitude of pharmacy students towards pharmacogenomics [12,13,14,15,16,17,18]. As per our knowledge, no such research/survey has been conducted so far in Riyadh to examine the impact of different attributes in knowledge and attitude towards pharmacogenomics in pharmacy students. Hence, the present study was designed to investigate the knowledge and attitudes regarding pharmacogenomics among pharmacy students at King Saud University, Riyadh.

## 2. Materials and Methods

### 2.1. Study Design, Settings

A cross-sectional study was carried out during the two months from 1 July to 31 August 2021. This study used a validated questionnaire. The targeted sample involved pharmacy students. The survey was designed according to the literature to assess the knowledge, views, attitudes, behavior, and interest in genomic medicine and PGX among health science students [4].

### 2.2. Questionnaire

The survey was distributed in English and involves 3 parts. Part 1 consisted of demographic data such as age, gender, pharmacy year, and education degree. Part 2 consisted of knowledge questions; it included 8 multiple choice questions about genomic medicine and PGX. Based on 6 true or false questions about genetics and 2 multiple-choice questions, a knowledge score was calculated. The knowledge score was calculated by counting all of the correct answers on the knowledge questionnaire; the maximum score was 8. According to the number of correct answers, 3 knowledge levels were created: high (7–8 correct answers), moderate (5–6 correct answers), and low (4 or fewer correct answers). Note that those who answered incorrectly and those who did not know the correct answers were combined in one group. Part 3 consisted of attitude questions and included 5 categories that were identified regarding the attitudes towards genomic medicine and PGX: views and considerations, desire to participate, and accessibility and availability of genetic tests. All the questions were scored on a 5-point Likert scale: strongly disagree, disagree, neutral, agree, and strongly agree. To facilitate analysis, strongly agree and agree merged as agree and strongly disagree and disagree merged as disagree.

### 2.3. Statistical Analysis

A descriptive analysis was conducted to assess the prevalence and sociodemographic factors of the study population. A Chi-square test was used for the categorical variables analysis whenever applied. The data was analyzed using Statistical Package for Social Sciences version 26.0 (SPSS Inc., Chicago, IL, USA), and a *p* value of <0.05 was considered statistically significant.

## 3. Results

### 3.1. Characteristics of Study Subjects

A total of 552 students were included in the current study and agreed to complete the questionnaire. Most of the students were aged between 18–25 years (516 of 552; 93.5%). Almost 97% (535 of 552) of included subjects were Saudi nationals. Only 31.7% (175 of 552) of pharmacy students had completed coursework related to pharmacogenomics and pharmacogenetics. Table 1 contains the demographics and other characteristics of the study subjects.

### 3.2. Assessment of Knowledge of Pharmacogenomics in Study Subjects

Table 2 displays the various questions used to assess the knowledge of pharmacogenomics and their responses by pharmacy students. Among the participants, 41.8% (231 of 552) correctly defined pharmacogenomics, and more than half (54.0%) knew the exact number of chromosomes in humans. A total of 449 participants (81.3%) knew that adverse reactions can be caused by genetic changes. A total of 58.2% of study subjects agreed that the FDA recommends pharmacogenomics testing for certain drugs. Most of the students didn’t know the year in which pharmacogenomics was used in clinical settings, with only 15.2% of students providing the correct answer.

### 3.3. Association of Pharmacogenomics Knowledge with Demographic and Academic Characteristics

Among female subjects, 4.5% had a high level of knowledge compared to 6.5% of males. Similarly, 32.0% females had a low level of knowledge compared to 19.4% of males. Overall, males had a mean knowledge score of 4.5 ± 1.5 compared to 4.1 ± 1.5 in females, and the difference was statistically significant (*p* = 0.011). There was a significantly higher degree of knowledge with each passing year of the degree course (*p* = 0.013). Among subjects who had completed coursework related to pharmacogenomics and pharmacogenetics, about 6.4% had a high level of knowledge compared to only 4.3% among those without any coursework in pharmacogenomics and pharmacogenetics. Similarly, among subjects who had completed coursework related to pharmacogenomics and pharmacogenetics, about 20.0% had a low level of knowledge compared to 31.9% of subjects who had not completed any coursework in pharmacogenomics and pharmacogenetics (*p* = 0.018). Overall, subjects with any sort of coursework in pharmacogenomics and pharmacogenetics showed a mean knowledge score of 4.6 ± 1.5 in comparison to 4.1 ± 1.6 in those subjects without any coursework in pharmacogenomics and pharmacogenetics (*p* = 0.018). Table 3 contains the association of level of knowledge with the other academic and demographic features of study subjects.

### 3.4. Attitude, Perception, and Thoughts of Study Subjects about Pharmacogenomics and Genomic Medicine

In order to assess the attitude, perception, and thoughts of study subjects on genomic medicine and pharmacogenomics, some questions were asked. When we asked if students would consider taking a genetic test to evaluate their risk of acquiring certain hereditary diseases in the future, 58.3% agreed while 32.2% disagreed and 9.4% were neutral. When asked, “if someone in their family was diagnosed with cancer, would they consider genetic testing as a means to choose a cancer treatment with fewer side effects,” 69.7% agreed, 16.3% disagreed, and 13.9% were neutral. Table 4 contains various questions to assess the attitude, perception, and thoughts of study subjects on genomic medicine and pharmacogenomics.

Table 4 shows the questions to assess the desire of pharmacy students to participate in pharmacogenetic and pharmacogenomic research. A total of 61.2% (338 of 552) wanted to attend a pharmacogenetics course or seminar, while 56.2% (310 of 552) wanted to participate in genetic research. The majority of students were opposed to genetic testing ordered online. Table 5 shows the availability and accessibility of genetic testing.

## 4. Discussion

The main objective of pharmacogenomics is to assess the impact of an individuals’ genetic makeup on drug efficacy and tolerance, and it is a vital factor in the new era of tailored medicine [18]. The field of pharmacogenomics is quite fresh in the Middle East. The present study was designed to estimate the degree of knowledge and attitude towards pharmacogenomics and pharmacogenomics of students pursuing pharmacy education at their bachelor’s and master’s levels. To the best of our knowledge, there are only a few awareness studies in students and health care professionals in Asia. Therefore, our study fills in an important gap in research on education and attitude toward pharmacogenomics and its implementation into clinical practice. A total of 452 students were enrolled in the study based on a questionnaire to be completed by participants regarding the various aspects of pharmacogenomics and pharmacogenetics. All the participants completed the prerequisite questionnaire for the conduct of study. In accordance with previous studies, the participants in our study were mostly female (70.3%), less than 30 years old (99.1%), and undergoing pharmacy training (100%) [19,20]. As all the students, especially Saudi nationals, enrolled in the study, it reflects the positive attitude of Arab students towards the study. The study group was different in a prior report published from Kuwait by Albassam et al. (2018) [21].

In addition, we demonstrated that majority of pharmacy students had a fair knowledge of pharmacogenomics, which was evident through a detailed questionnaire. These findings are in line with the studies from UK [22] and USA [23]. Our observations are in accordance with the study of [24], who reported that the bulk of pharmacy students from eight different pharmacy colleges in California had information about the subject of pharmacogenomics and approved of it as career specialization. The best application of pharmacogenomics is understanding relevant medication-gene pairing and pharmacists are primed to help providers recognize when such an important pairing exits in a medication they are prescribing.

According to our study, male participants have a significantly higher level of knowledge of pharmacogenomics when compared to female participants, which is in contradiction with the survey conducted by Bagher et al. who did not observe any significant difference in the knowledge of pharmacogenomics between males and females [25]. We observed a significantly increasing knowledge score with every passing year of a pharmacy degree course, which is evident due to the fact that with each year of course, students are increasingly exposed to higher and exhaustive study material regarding the subject and receive a chance to apply their knowledge in practice in order to enhance their skills. Individuals who had completed additional coursework in pharmacogenomics and pharmacogenetics had a significantly higher level of knowledge of the subject than those without any additional coursework, which is in agreement with the observations of Bagher et al. who observed a higher degree of knowledge in students with higher degrees and additional coursework [25]. In countries like Canada, the UK, France, the United States, and Saudi Arabia, continuing education on pharmacogenomics and pharmacogenetics has become mandatory for not only renewing but sustaining a pharmacist’s license [26,27]. Contrary to this, in Kuwait, continuing education programs for pharmacists are not obligatory [28]. The continuing education programs on pharmacogenomics and pharmacogenetics will improve pharmacists’ knowledge about the subject and will assist in the appropriate delivery of pharmacogenetics testing [25]. We did not find any significant difference in knowledge between Bpharm and PharmD students, which is in contrast to earlier studies that observed a statistically higher knowledge score of pharmacogenomics and pharmacogenetics in PharmD students compared to Bpharm students. In contrast, [29] reported that pharmacists with Bpharm degrees earned considerably lower knowledge scores than pharmacists with PharmD degrees. The variation in the two studies could be due to the fact that King Saud university runs a comprehensive pharmacology program. Above all, undergraduate pharmacy students enrolled in PharmD or Bpharm programs at this university are educated in basic concepts in pharmacogenomics and pharmacogenetics. Therefore, the undergraduate as well as post-graduate pharmacy students have the skills and knowledge to implement pharmacogenetics testing at the clinical level. In fact, the number of pharmacy lectures is higher than that recommended by the International Society of Pharmacogenomics [30].

One of the unique characteristics of our study is that it is the first attempt in Saudi Arabia to investigate students’ attitudes and perceptions towards genomic personalized medicine and pharmacogenomics as well as their social, ethical, and legal implications. The knowledge and thoughts were assessed through various questions, and it was observed that majority of the students agreed upon the applicability and positive role of genomic medicine in efficient clinical practice and patient care. Moreover, 60% of the respondents agreed with the following statements: Pharmacogenetics is important; In the future, respondents might consider a genetic test to determine their risk of developing certain genetic diseases; Respondents were only interested in knowing their vulnerability to diseases that have existing interventions for protection; When facing a cancer diagnosis, respondents would consider genetic testing as a means to choose a cancer treatment with fewer side effects; If respondents family has a history of diabetes, they would consider having their genes analyzed for choosing a treatment with minimal adverse effects; Their physician should explain their genome report to respondents; and Their pharmacist should also be able to explain their genome. In particular, the results indicate a moderate confidence regarding respondents’ familiarity with the term “pharmacogenomics” and its implementation in personalized medicine. To enhance the practice of pharmacogenomics in personalized medicine, the knowledge of pharmacy students on relevant topics should be improved, as also indicated in earlier published reports [31,32]. To support this thought, a previous study in San Francisco advocated for the feasibility and importance of educating future pharmacists by incorporating pharmacogenetic testing into professional school curricula [33].

This study was carried out in the Riyadh region of Saudi Arabia, thus, generalizing the outcome of this survey to other locations may not provide accurate representation. Moreover, an extensive survey to accommodate the participation of pharmacists with different experiences and from all the regions of Saudi Arabia is recommended. Future studies could aim at including a higher number of participants to further validate the present observations, while also including students from other pharmacy-related schools as bigger target groups. 

## 5. Conclusions

Modern genomic technologies and their scope for use in clinical practice have not only revolutionized the oncology health sector, but are becoming more relevant in fields such as psychiatry and neurology. As a result, there is a need to create awareness among pharmacy students to overcome the knowledge gap in pharmacogenomics. Our study could help to explore the most effective and feasible approaches to improve students’ awareness and knowledge about pharmacogenomics and pharmacogenetics. Integrating full-time pharmacogenomics courses into the pharmacy curriculum, continuous training modules, etc. are some of the vital steps that need to be undertaken. Pharmacogenomics is exciting and certainly a focus of much research to come. The incorporation of pharmacogenomics and pharmacogenetics in clinical practice may predict treatment outcomes, decrease adverse drug effects, and have a positive impact on the future of patient care.

## Figures and Tables

**Table 1 genes-13-00269-t001:** The participants’ demographic characteristics.

Variables	*n* (%)
Gender	
Male	164(29.7)
Female	388(70.3)
Age group	
<18	11(2.0)
18–25	516(93.5)
26–30	20(3.6)
31–35	2(0.4)
>36	3(0.5)
Pharmacy year	
First	83(15.0)
Second	98(17.8)
Third	88(15.9)
Fourth	72(13.0)
Fifth	82(14.9)
Final	129(23.4)
Nationality	
Saudi	535(96.9)
Non Saudi	17(3.1)
Parents work in health care setting	
Yes	104(18.8)
No	448(81.2)
Coursework related to Pharmacogenomics and Pharmacogenetics	
Yes	175(31.7)
No	377(68.3)

**Table 2 genes-13-00269-t002:** The knowledge of pharmacogenomics among pharmacy students.

Questions	Correct Answer *n* (%)	Vague Answer *n* (%)
Definition of pharmacogenomics	231(41.8)	321(57.8)
Humans have 48 chromosomes	298(54.0)	254(46)
Adverse reactions can be caused by genetic changes	449(81.3)	103(18.7)
FDA recommends pharmacogenomics testing for certain drugs	321(58.2)	231(41.8)
Certain drugs can be affected by genetic changes in the patient	432(78.3)	120(21.7))
Several environmental factors can affect gene expression, including tobacco smoke	345(62.5)	207(37.5)
Which year PGx has been used in clinical settings?	84(15.2)	468(84.8)
Which of the following groups was established to create peer-reviewed, evidence-based clinical guidelines for specific gene-drug pairs	184(33.3)	386(66.7)

**Table 3 genes-13-00269-t003:** A comparison of the level of knowledge by demographic and academic characteristics.

Level of Knowledge					
	Mean Score (SD)	Low *n* (%)	Moderate *n* (%)	High *n* (%)	*p* Value
Total	4.2(1.6)	153(27.7)	361(65.4)	27(4.9)	
Gender					0.011
Male	4.5(1.5) *	31(19.4)	119(74.4)	10(6.3)	
Female	4.1(1.5)	122(32.0)	242(63.5)	17(4.5)	
Age group					0.0624
<18	3.5(2.1)	6(54.5)	4(36.4)	1(9.1)	
18–25	4.3(1.6)	140(27.7)	341(67.4)	25(4.9)	
26–30	4.5(1.8)	5(26.3)	13(68.4)	1(5.3)	
31–35	3.0(1.4)	1(50)	1(50)	--	
>36	4.3(2.0)	1(33.3)	2(66.7)	--	
Pharmacy year					0.013
First	3.7(1.7) *	31(38.8)	47(58.8)	2(2.5)	
Second	4.0(1.6)	36(37.5)	56(58.3)	4(4.2)	
Third	3.9(1.6)	26(31.0)	55(65.5)	3(3.6)	
Fourth	4.3(1.5)	21(29.2)	48(66.7)	3(4.2)	
Fifth	4.5(1.4)	20(24.4)	57(69.5)	5(6.1)	
Final	4.8(1.4)	19(15.0)	98(77.2)	10(7.9)	
Education degree					0.931
Bachelors	4.2(1.6)	107(28.2)	255(67.1)	18(4.7)	
Pharm D	4.3(1.6)	46(28.6)	106(65.8)	9(5.6)	
Parents work in health care setting					0.088
Yes	3.9((1.7) *	37(37.4)	58(58.6)	4(4.0)	
No	4.3(1.5)	116(26.2)	303(68.6)	23(5.2)	
Course related to Pharmacogenomics and genetics					0.018
Yes	4.6(1.5) *	35(20.5)	125(73.1)	11(6.4)	
No	4.1(1.6)	118(31.9)	236(63.8)	16(4.3)	
Nationality					1.0
Saudi	4.3(1.6)	149(28.3)	351(66.7)	26(4.9)	
Non-Saudi	3.8(2.1)	4(26.7)	10(66.7)	1(6.7)	

* significant *p*-value < 0.05.

**Table 4 genes-13-00269-t004:** The attitudes, thoughts, and desire to participate in genomic medicine and PGX research.

Questions	Agree *n* (%)	Neutral *n* (%)	Disagree *n* (%)
In the future, I might consider a genetic test to determine my risk of developing certain genetic diseases	332	52	178
	(58.3)	(9.4)	(32.2)
I am only interested in knowing my susceptibility to diseases that have current interventions for protection	291	59	202
	(52.7)	(10.7)	(36.6)
When facing a cancer diagnosis, I would consider genetic testing as a means to choose a cancer treatment with fewer side effects	385	77	90
	(69.7)	(13.9)	(16.3)
If my family has a history of diabetes, I would consider having my genes analyzed so that I can choose a treatment with minimal adverse effects	323	69	160
	(58.5)	(12.5)	(29.0)
My physician should explain my genome report to me	358	144	50
	(64.9)	(26.1)	(9.1)
My pharmacist should explain my genome report to me	326	170	56
	(59.1)	(30.8)	(10.1)
I am interested in attending a genetics (PGX) course or seminar	338	163	51
	(61.2)	(29.5)	(9.2)
Interested in having my genetics collected by a biobank	443	-	109
	(80.3)	-	(19.7)
I am interested in participating in genetic research	310	171	71
	(56.2)	(31.0)	(12.9)

**Table 5 genes-13-00269-t005:** The accessibility and availability of genetic testing.

Questions	Agree *n* (%)	Neutral *n* (%)	Disagree *n* (%)
Insurance companies and future employers are unable to perform genetic tests	222(40.2)	260 (47.1)	70(12.7)
My family has a history of serious genetic diseases, so I am glad that genetic tests are available to help people determine if they are at risk	409(74.1)	109(19.7)	34(6.2)
Genetic tests can now be ordered online, which is great	304(55.1)	168(30.4)	80(14.5)

## Data Availability

The datasets used and analyzed during the current study are available from the corresponding author on reasonable request.

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
