# Peer review of "Knowledge, Attitude and Perception of Pharmacy Students towards Pharmacogenomics and Genetics: An Observational Study from King Saud University"

_genes, 2022, doi:10.3390/genes13020269_

Round 1
Reviewer 1 Report
These authors have performed a study examining pharmacy students’s knowledge, attitude and perception of pharmacogenomics.
- For background purposes, it would be important to know how prevalent pharmacogenomic testing is being conducted in Saudi Arabia, notably the Riyadh region. This will make this data more comparable to other regions of the world where these surveys have been conducted. If they are not readily available, the value of pharmacogneomics an interest in adopting is diminished.
- Table 2. How do the authors explain the majority of participants didn’t know how many chromosomes are present in humans? Or was this a typo, and the question should have been 46?
- Table 2 asks which year PGx has been used in clinical setting is difficult to answer correctly. What defines first clinical use?
- Table 3. In evaluating performance vs. pharmacy year, what year is PGx typically offered?
- A very similar study was published in 2017 and should be discussed and referenced. J Pharm Educ 2017;8(1): Article 2. http://pubs.lib.umn.edu/innovations/vol8/iss1/2.

Author Response
- For background purposes, it would be important to know how prevalent pharmacogenomic testing is being conducted in Saudi Arabia, notably the Riyadh region. This will make this data more comparable to other regions of the world where these surveys have been conducted. If they are not readily available, the value of pharmacogenomics an interest in adopting is diminished.
Response:
Response: In Saudi Arabia the prevalence of pharmacogenomics testing (CYP2C9*2 and CYP2C9*3) were found with rates of 11% and 9%, respectively [11]. Additionally, studies also found that genetic variation in Saudi Arabia is high due to some reasons, (cousin marriages 50%) which makes important for educating the pharmacist about pharmacogenomics [11,12]. There have been several reports published previously from around the world to study the knowledge or attitude of pharmacy students towards pharmacogenomics
- Table 2. How do the authors explain the majority of participants didn’t know how many chromosomes are present in humans? Or was this a typo, and the question should have been 46?
Response: we apologies for the confusion. This question was asked on 3 option scale among those one of the answer is correct and the remaining 2were vague. The correct answer of it is 46
- Table 2 asks which year PGx has been used in clinical setting is difficult to answer correctly. What defines first clinical use?
According to literature it was came into effect in 2010 (Saudi food drug authority was first approved in clinical setting), we defined it as first clincial testing when the patient who has taking certain antiepileptic drugs, they got hypersensitivity reactions, in this clincial case application of PGx was help full to reduce this incidence
Response:
- Table 3. In evaluating performance vs. pharmacy year, what year is PGx typically offered?
Pharmacogenomics was in general firstly introduced by SFDA in clincial practice was in 2010 in Saudi Arabia, while with respect to academics it was implemented more recently in 2022. Although 3rd and 4 year of pharmacy students covered some of the topics in their subjects during their graduation.
Response:
- A very similar study was published in 2017 and should be discussed and referenced. J Pharm Educ 2017;8(1): Article 2. http://pubs.lib.umn.edu/innovations/vol8/iss1/2.
Response: The article has been cited in the discussed and properly cited in the revised manuscript now.
Reviewer 2 Report
DearAuthors,
Pharmacogenetics is a rapidly growing field studying of how genetic differences influence the variability of individual patient responses to drugs, aims to distinguish responders from non-responders and predict those in whom toxicity and It can be regarded as the 21st century's answer for the rational use of drugs - the right drug to the right patient at right dose. To this end, it is understandable that pharmacogenomics is expected to play a large role in personalized medicine. It is also increasingly being explored in the primary care context.
The most important barriers delaying clinical uptake and application of pharmacogenomics is lack of knowledge and insufficient education of health professionals regarding pharmacogenetics and genomics rather than technical issue. Since the integration of pharmacogenetics (PGx) testing into routine care will, in part, depend upon the physicians’, pharmacists’ and as well as patients’ acceptance the education of scientists, healthcare professionals and publics in genetics is crucial for appropriate application of pharmacogenetics to integrate into healthcare system. . Understanding of implication of personalized medicine and so pharmacogenetics in drug reactions incidence will allow healthcare professionals to be an integral part of the new age of personalized medicine. Arafah and his colleague’s by means of cross-sectional survey study aimed to explore how well Pharmacy students in Saudi Arabia perceive importance and application of genomic medicine and pharmacogenetics. To this end, this manuscript highlights the potential opportunities and applications of PGx in clinical practice pf pharmacy. Over all nice job! The manuscript is very well written and the introduction provides a good, generalized background about the importance of PGx
Thanks
Author Response
Reviewer#2
Pharmacogenetics is a rapidly growing field studying of how genetic differences influence the variability of individual patient responses to drugs, aims to distinguish responders from non-responders and predict those in whom toxicity and It can be regarded as the 21st century's answer for the rational use of drugs - the right drug to the right patient at right dose. To this end, it is understandable that pharmacogenomics is expected to play a large role in personalized medicine. It is also increasingly being explored in the primary care context.
The most important barriers delaying clinical uptake and application of pharmacogenomics is lack of knowledge and insufficient education of health professionals regarding pharmacogenetics and genomics rather than technical issue. Since the integration of pharmacogenetics (PGx) testing into routine care will, in part, depend upon the physicians’, pharmacists’ and as well as patients’ acceptance the education of scientists, healthcare professionals and publics in genetics is crucial for appropriate application of pharmacogenetics to integrate into healthcare system. Understanding of implication of personalized medicine and so pharmacogenetics in drug reactions incidence will allow healthcare professionals to be an integral part of the new age of personalized medicine. Arafah and his colleague’s by means of cross-sectional survey study aimed to explore how well Pharmacy students in Saudi Arabia perceive importance and application of genomic medicine and pharmacogenetics. To this end, this manuscript highlights the potential opportunities and applications of PGx in clinical practice pf pharmacy. Over all nice job! The manuscript is very well written and the introduction provides a good, generalized background about the importance of PGx
Greetings, reviewer First and foremost, thank you for your feedback. We completely agree with your remark that the most significant barrier to PGx in clinical practice is a lack of information and education. As a result, there is a need to raise awareness among pharmacist in order to close the pharmacogenomics knowledge gap.
Reviewer 3 Report
Important study as physicians often consult with pharmacists regarding medication choice and potential side effects.
A few comments:
The introduction needs editing in line 36 to better convey the role of pharmacogenomics and possibly introduce the term pharmacogenetics. It is the study of how genetic variations may influence medication efficacy as well as tolerance. Sentence 35-36 does not convey that adequately.
Line 44 misspells PharmGKB.
The term personalized medication sometimes is too overreaching. Explaining that a bit better and with more validity would be helpful.
Similarly, the discussion section needs more clarity as line 143-144 sounds too hyperbolic and the term decipher does not seem quite right here. Line 150-151 also needs clarification; there is a repeat of pharmacogenomics twice and it does not read well.
There should be mention that the best application of PGx is understanding relevant medication-gene pairing and pharmacists are primed to help providers recognize when such an important pairing exits in a medication they are prescribing.
The study itself is well done. Would just like to see more work put into the introduction and discussion to be more relevant and precise.
Under conclusions, the authors should be more specific about which areas of medicine have been "revolutionized". That is a strong word. I suspect the work in oncology has been revolutionary to some extent but in other fields like psychiatry it is becoming more relevant but not revoluationary.
The authors need to be careful about the over exaggerations. They should stick to the studies and the facts without too much hyperbole. "Certainly improve outcome..." ? (line 232) Another strong statement that needs to be clarified and seems over stated. Again, PGx is exciting and certainly a focus of much research to come, but the authors need to be careful of statements that are too powerful without specific examples.
Author Response
- The introduction needs editing in line 36 to better convey the role of pharmacogenomics and possibly introduce the term pharmacogenetics. It is the study of how genetic variations may influence medication efficacy as well as tolerance. Sentence 35-36 does not convey that adequately.
Response: The lines have been rephrased to accommodate the word pharmacogenetics and now the sentence conveys the message clearly
- Line 44 misspells PharmGKB.
Response:
- The term personalized medication sometimes is too overreaching. Explaining that a bit better and with more validity would be helpful.
Response: The sentence has been changed to make it more valid and repress the context of the sentence.
- Similarly, the discussion section needs more clarity as line 143-144 sounds too hyperbolic and the term decipher does not seem quite right here. Line 150-151 also needs clarification; there is a repeat of pharmacogenomics twice and it does not read well.
Response: Line 143-145 has been rephrased. The word pharmacogenomics has been deleted and the sentence (line 150-151) changed accordingly.
- There should be mention that the best application of PGx is understanding relevant medication-gene pairing and pharmacists are primed to help providers recognize when such an important pairing exits in a medication they are prescribing.
Response: This sentence has been incorporated in the discussion section now
- The study itself is well done. Would just like to see more work put into the introduction and discussion to be more relevant and precise.
Response: The introduction and discussion section have been modified and updated as per comments and suggestion of reviewers
- Under conclusions, the authors should be more specific about which areas of medicine have been "revolutionized". That is a strong word. I suspect the work in oncology has been revolutionary to some extent but in other fields like psychiatry it is becoming more relevant but not revolutionary.
Response: The desired changes have been made in the conclusion now.
- The authors need to be careful about the over exaggerations. They should stick to the studies and the facts without too much hyperbole. "Certainly improve outcome..." ? (line 232) Another strong statement that needs to be clarified and seems over stated. Again, PGx is exciting and certainly a focus of much research to come, but the authors need to be careful of statements that are too powerful without specific examples.
Response: The points have been noted and the conclusion has been made more modest.